# The Role of Personality Traits, Cooperative Behaviour and Trust in Governments on the Brexit Referendum Outcome

Francisco J. Areal 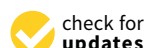

Centre for Rural Economy, School of Natural and Environmental Sciences, Newcastle University, Newcastle upon Tyne NE1 7RU, UK; Francisco.Areal-Borrego@newcastle.ac.uk

**Abstract:** We analyse the role of personality traits along with individuals' cooperative behaviour, level of trust in the UK government and the European Council (EC, the body that defines the European Union's overall political direction and priorities) and socio-demographics on UK citizens' voting choices on the 2016 Brexit referendum. We use data from a survey conducted in April 2019 on 530 UK citizens who voted in the 2016 Brexit referendum. We use a Probit model to investigate what role voters' personality traits, their trust in government institutions, their level of cooperative behaviour and socio-demographics played in the way they voted. We find voters' choice was associated voters' personality traits. In particular, voters associated with being extraverted, acting with self-confidence and outspokenness (i.e., agency), and voters' closeness to experience, to forming part of a diverse community and the exchange of ideas and experiences were found to be associated with voting for Brexit in the 2016 referendum. We found that voters' willingness to cooperate with others was associated with being less likely to vote for Brexit. In addition, voters who trusted the UK government were more likely to vote for Brexit, whereas voters trusting the EC were more likely to vote for the UK to stay in the EU. We also found that voters with relatively high level of education were less likely to vote for Brexit and voters not seeking jobs were more likely to vote for Brexit than students, unemployed and retired. We conclude that incorporating personality profiles of voters, their pro-social behaviour as well as their views on trust in politicians/government institutions, along with socio-demographic variables, into individuals' vote choice analysis can account for voter heterogeneity and provide a more complete picture of an individual's vote choice decisions, helping to gain a better understanding of individual vote choices (e.g., better predictions of future individual vote intentions).

**Keywords:** Brexit; personality traits; cooperative behaviour; referendum; vote choice

## 1. Introduction

Brexit and its consequences remain a dominant political and economic affair in both international and domestic policy in the UK and the EU. On 23 June 2016, the United Kingdom voted to leave the European Union (EU) membership on a public referendum with a 72.2% turnout. UK citizens were asked "Should the UK remain a member of the European Union or leave the European Union?" and 51.9% (17,410,742) of voters in the UK voted to leave vs. 48.1% (16,141,241) who voted to stay within the EU.

Brexit has been analysed from a wide range of angles including identifying reasons for the results of the referendum (Alabrese et al. 2019; Arnorsson and Zoega 2018; Goodwin and Heath 2017); understanding how uncertainty associated with the Brexit referendum result may affect firms and industries (Hill et al. 2019); the effect of Brexit on EU and UK stock markets and exchange rates (Bashir et al. 2019; Dao et al. 2019; Guedes et al. 2019; Shahzad et al. 2019); the effects of Brexit on trade (Dhingra et al. 2017); the effects of Brexit on specific industries such as agriculture and seafood (Symes and Phillipson 2019); the potential impact of Brexit on the Pelagic Advisory Council to secure a long-term sustainable pelagic fisheries management (Ohms and Raakjær 2019); and how Brexit may affect the health services in the UK (Fahy et al. 2019).

Understanding why Brexit happened and its implications gained attention before and after the referendum was held, and this continues to be the case. Here, we aim to gain understanding of the reasons behind the Brexit result. This is important for predicting post-Brexit policies in the UK and helping to ascertain whether other EU countries may decide to leave the EU (Hobolt and Rodon 2020).

Whereas early research on identifying reasons for the results of the 2016 referendum focused on socio-demographic factors such as education profile, age, employment and life satisfaction (Alabrese et al. 2019; Arnorsson and Zoega 2018; Goodwin and Heath 2017), political psychology and economic research highlighted the importance of personality traits in explaining political behaviour and economic behaviour (Fatke 2017; Drouvelis and Georgantzis 2019). In addition, it has been recommended that economic analysis look beyond the socio-demographic and financial aspects. In particular, Drouvelis and Georgantzis (2019) advocate for incorporating the role of personality traits into economic analysis; Wang (2016) pointed out the insufficiency in the study of personality traits and voting behaviour and advocated for further research in this area; and earlier, Caprara et al. (1999) and Caprara and Zimbardo (2004) highlighted the importance of understanding the association between voters' personal characteristics, political agendas and political choice. Despite these recommendations and findings that personality variables may be more important in explaining variance of voting intentions than socio-demographic variables (Barbaranelli et al. 2007), little attention has been paid to analysing the relationship between personality traits and individuals' vote choices (Caprara et al. 1999; Blais and St-Vincent 2011). Garretsen et al. (2018) and more recently May et al. (2021) also used behavioural approaches to analyse the 2016 Brexit referendum vote. Garretsen et al. (2018) and Alabrese et al. (2019) used a regional level-based approach as opposed to the individual level-based approach used here. However, whereas Alabrese et al. (2019) found that voters' education profiles, their historical dependence on manufacturing employment as well their level of income were determinants of how local authority areas voted in the 2016 referendum, Garretsen et al. (2018), using a regional clustering of personality traits, found that voters' personality traits explained the regional dispersion of the 2016 referendum. More specifically, personality traits such as openness, agreeableness and conscientiousness were associated with the Brexit vote, being openness the most important personality trait. Voters' perceptions and attitudes towards the EU have also been found important in explaining the vote. May et al. (2021) analysed how farmers' perceptions and attitudes towards the EU as well as farmers' perceived capacity to control factors that impact farm performance influenced the way they voted. Human values have been also found to influence European identity, views on immigration and trust in politicians; the former two were more important in explaining the voting choice in the Brexit referendum than socio-demographic characteristics (Dennison et al. 2020).

Taking into consideration that political trust and individual's intrinsic motives such as cooperative behaviour may have been associated with the way people voted in the referendum (Abrams and Travaglino 2018; Liberini et al. 2019), we extend earlier approaches used to investigate whether aspects such as an individual's personality traits, trust in governmental institutions and cooperative behaviour along with their socio-demographic characteristics are associated with the way people voted in the 2016 referendum.

We contribute to this literature by analysing whether the role of individual level psychological traits and cooperative behaviour along with level of trust in government institutions, including the UK government and the European Council, the body that defines the European Union's overall political direction and priorities, are associated with the 2016 Brexit referendum outcome. To the best of our knowledge, we are the first paper to combine these three elements to explain voting choice. We are not aware of previous research incorporating individuals' cooperative behaviour into voting choice. In addition, we investigate the 2016 referendum vote at the individual/voter level, which differs from analysis by Garretsen et al. (2018), which is conducted at the regional level (e.g., using regional personality traits). In other words, the research questions are whether an

individual's (a) personality traits; (b) political trust; and (c) cooperative behaviour are associated with the way they voted in the 2016 Brexit referendum when controlling for socio-demographics.

A better understanding of the linkages between these aspects and voter choices is needed to explain voting preference heterogeneity. Understanding the sources of heterogeneity on how people vote can have implications for policy development and analysis.

## 2. Materials and Methods

### 2.1. Conceptual Framework

In this section, we present the mechanisms through which different elements are considered to be linked to people's voting choices regarding the 2016 referendum. Incorporating a psychological perspective has been pointed out to be lacking when explaining the Brexit vote (Garretsen et al. 2018). Geographical psychology research has found an association between regional differences in personality traits and regional socio-economic indicators (Rentfrow et al. 2015; Rentfrow 2010; Rentfrow et al. 2008). This research has served as a basis to account for spatial heterogeneity personality traits in explaining the Brexit vote (Garretsen et al. 2018). We take this further by accounting for individual level heterogeneity in personality traits to investigate the association between individual level personality traits and Brexit vote. Voting for either leaving the EU or staying in the EU are hypothesised to be dependent on aspects such as a voter's psychological traits, their level of trust in government institutions and their cooperative behaviour, along with their socio-demographic characteristics. Regarding the voter's psychological traits, it is hypothesised that voters with a high level of agency will be more likely to vote for leaving the EU in the 2016 referendum. These voters showing the trait of agency are often associated with dominant characteristics, acting with self-confidence and outspokenness. We expect that self-confident individuals may be more likely to support Brexit since they may believe the UK will be able to succeed outside the EU. Agreeableness captures the good naturedness, compassionateness, cooperativeness and trustworthiness of individuals. Hence, a priori, we do not have any strong expectation on how agreeableness and voting choice may be associated. Personality characteristics such as originality, curiosity and ingenuity belong to the openness to experience trait. This may be related to voter's receptivity to new ideas and experiences, which may be associated with forming part of a diverse community. Hence, we hypothesise that voters with an openness to experience trait would be more likely to vote for the UK staying as part of the EU. Neuroticism is associated with the lack of emotional stability and lack of negative emotions by individuals. A priori, we do not foresee any association between neuroticism and voting choice. Regarding extraverted individuals, those who are talkative, sociable, assertive, enterprising and energetic, it is unclear whether they would be more inclined to vote for the UK to leave or stay in the EU. Extraverted individuals might tend to search for novel experiences (i.e., UK leaving the EU) (Gocłowska et al. 2019), but also enjoy being with other people, indicating that they might tend to vote for the UK to stay. Finally, the conscientiousness trait captures an individual's personality characteristics, such as orderliness, responsibility, self-discipline and dependability, and is not expected to be related to either vote choice.

The second main aspect that we include in the analysis if the level of trust that voters/citizens have in government institutions. Trust is a key aspect in understanding how individuals interact among themselves and with institutions (including government institutions) (Abrams and Travaglino 2018; Levi and Stoker 2000). Trust in politicians has been found to be positively associated with voting to remain in the EU (Dennison et al. 2020; Hobolt 2016). Hence, we hypothesise that the level of trust in government institutions is associated with the way voters voted for in the 2016 EU referendum. We expect that individuals who trust the UK government would be more likely to vote for the UK to leave the EU and individuals trusting the EC would be more likely to vote for the UK to stay in the EU.

The rationale for human cooperation is variable (Nettle et al. 2011). Thus, individuals may act cooperatively because there is an expectation of a future payment, but cooperation also can occur without expecting any compensation for it (Fehr and Gächter 2000). We hypothesise that people more willing to cooperate were more inclined to vote to remain in the EU than people who voted to exit the EU. This may be the case if individuals with a cooperative vision may perceive cooperation between countries to lead to positive outcomes for all as opposed to individuals with a self-interest vision where they perceive one country (e.g., the UK) would be better off making its own decisions. We make use of a public good game to elicit individuals' level of cooperativeness (Fehr and Gächter 2000). Public goods games have been previously used to account for an individual's willingness to cooperate with other individuals. For instance, Tsusaka et al. (2015) used a public good game to investigate how cooperative behaviour of neighbours influences an individual's own cooperative behaviour.

Figure 1 shows the conceptual framework and the elements that may be associated with the way individuals voted in the 2016 Brexit referendum.

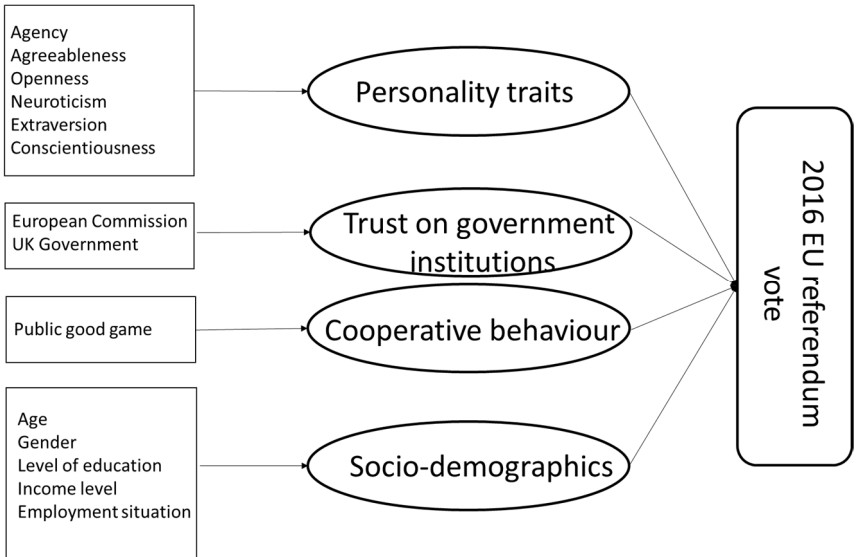

**Figure 1.** Conceptual framework.

### 2.2. Survey

We investigate the association of voters' psychological traits, their level of trust in UK and EC government institutions and their willingness to cooperate, along with their socio-demographic characteristics, with their probability of voting for the UK leaving the EU in the 2016 Brexit referendum. For this, we used a cross-sectional survey to collect data on the elements highlighted in Table 1.

We used a cross-sectional online survey hosted and designed via Qualtrics® to collect information on the way people voted in the referendum along with elements highlighted in Figure 1. Qualtrics used a panel with sampling quotas for the general population's age, gender and region (England, Wales, Scotland and Northern Ireland). Qualtrics invited respondents in the UK to complete the online survey in return for incentives/cash. The sample used a total of 631 individuals who took part in the survey conducted from 1 April to 10 April 2019, of which 544 voted in the referendum, 55 decided not to vote, 20 stated they could not vote and 12 did not answer. We used 530 valid responses from the survey to conduct the regression analysis. The survey was conducted prior to when a 6-month extension to ratify the Brexit withdrawal agreement was agreed between the UK and the EU27, until 31 October 2019. The questionnaire included sections on respondents' personality traits, level of trust in government institutions, cooperative behaviour (this

section consisted of a public good experiment, i.e., respondents playing the public goods game), a 2016 referendum vote choice question and socio-demographic variables.

### 2.2.1. Personality Traits

Individuals' patterns of behaviour, feelings and thoughts are associated with their personality. Although there are multiple combinations of traits that lead to unique personalities, studies of personality use questionnaire approaches to summarise major individual differences through the identification of personality dimensions (Caprara et al. 1999). In order to account for an individual's personality traits, the Midlife Development Inventory (MIDI) was used. The MIDI includes 6 personality trait scales: agency, agreeableness, openness, neuroticism, extraversion to experience sand conscientiousness (communion), and 30 self-ratings adjectives (Lachman and Weaver 1997). We used MIDI since it extends the Big 5 personality traits by including agency to identify individual's dominance and aggression characteristics. Each participant gave self-descriptions of their own personality on a 1 to 4 scale, ranging from not at all to a lot, using a list of 30 adjectives (Lachman and Weaver 1997). The 6 individual level personality traits were calculated following Lachman and Weaver (1997). The 6 traits were measured using self-ratings of 30 adjectives mainly selected from existing trait lists of inventories (Lachman and Weaver 1997). Respondents had to indicate how well each of the adjectives described them, from "a lot" (1) to "not at all" (4). Table 1 shows the 30 adjectives used to compute the personality trait scales. All adjectives are reverse scored, except "calm" and "careless".

**Table 1.** MIDI adjectives.

| Letter Code | Adjective | Letter Code | Adjective |
|:---:|:---:|:---:|:---:|
| a | Outgoing | p | Hardworking |
| b | Helpful | q | Imaginative |
| c | Moody | r | Soft-hearted |
| d | Organised | s | Calm |
| e | Self-confident | t | Outspoken |
| f | Friendly | u | Intelligent |
| g | Warm | v | Curious |
| h | Worrying | w | Active |
| i | Responsible | x | Careless |
| j | Forceful | y | Broad-minded |
| k | Lively | z | Sympathetic |
| l | Caring | aa | Talkative |
| m | Nervous | bb | Sophisticated |
| n | Creative | cc | Adventurous |
| o | Assertive | dd | Dominant |

In order to compute the 6 personality traits, the following calculations are performed (Lachman and Weaver 1997):

$$Agency = mean(e, j, o, t, dd) \tag{1}$$

$$Agree = mean(b, g, l, r, z) \tag{2}$$

$$Open = mean(n, q, u, v, y, bb, cc) \tag{3}$$

$$Nurot = mean(c, h, m, s) \tag{4}$$

$$Extrav = mean(a, f, k, w, aa) \tag{5}$$

$$Consc = mean(d, i, p, x) \tag{6}$$

### 2.2.2. Trust in Government Institutions

In order to capture a voter's trust in the UK government and the European Council, respondents were asked to evaluate the following two statements from strongly disagree

(1) to strongly agree (5): "I have trust in the UK government" and "I have trust in the European Council" (the body that defines the European Union's overall political direction and priorities).

### 2.2.3. Cooperative Behaviour: The Public Goods Name

In order to capture individuals' cooperative behaviour, we used a public goods game. This is a standard game in experimental economics to evaluate an individual's level of cooperation. During the game, each participant chooses how much they are willing to contribute to a common good. The following text was presented to respondents: "Imagine that you are part of a group of 4 people. Each group member receives GBP00 and has the possibility of increasing the group's earnings by giving a monetary contribution from the GBP100 received.

The total contributed by all members is doubled and then shared evenly by all members regardless of how much each member contributed. Money that an individual does not contribute to the group is kept by the individual. Hence, the total amount received by each member would be the money kept by the member plus double the total contribution by the group divided by 4 members."

Then, respondents were asked to answer: "How much would you contribute to the group of the GBP100 received?" We used the answer given to this question by respondents, which can vary between GBP0 to GBP100, as an indicator of cooperation behaviour.

### 2.2.4. 2016 Referendum Vote

The question on how the participants voted in the 2016 Referendum was "What did you vote on the Referendum on the UK membership of the European Union back in 2016?" Respondents could select: (1) Remain a member of the European Union; (2) Leave the European Union; (3) I decided not to vote; (4) I could not vote; or (5) Prefer not to say. We only used observations where individuals answered either that they voted (1) Remain a member of the European Union or (2) Leave the European Union, which accounted for 86% of respondents.

### 2.2.5. Socio-Demographic Characteristics

We included as socio-demographic characteristics age, gender, income level (less than GBP 20,000; between GBP 20,000 and GBP 39,999; between GBP 40,000 and GBP 59,999; GBP 60,000 or more), level of education (secondary school; college; university) and employment situation (working; student; retired; unemployed—seeking work; not in paid employment—not seeking job).

### *2.3. Econometric Analysis*

Based on the additive random utility model, we assume that an individual's $i$ utility for an alternative $j$, $U_{ij}$, is determined by a deterministic and an unobserved component, $V_{ij}$, and a random variable, $\varepsilon_{ij}$, accounting for effects on preferences of unobserved attributes of the alternative. In our case, the alternatives are two ($J = 2$), voting for the UK to remain in the EU and the UK to exit the EU.

$$U_{ij} = V_{ij} + \varepsilon_{ij} \tag{7}$$

where $V_{ij} = x'_{ij}\beta$ covariates are denoted as $x_i$ and $\beta$ is a vector of parameters associated with covariates $x_i$. We model the problem as the probability that the decision of individual $i$ on voting for the UK to leave (stay) in the EU is alternative $j$, conditional on the covariates $x_i$:

$$Pr(y_i = 1) = F_j(x_i, \theta) \tag{8}$$

where $y_i$ is the individual $i$ ($i = 1, \ldots, N$) decision on the referendum vote $j$ out of the $J = 2$ alternatives ($j = 1, 2$).

With respect to the covariates used, we include all elements included in the conceptual framework (Table 1). Therefore, we included psychological traits, trust in government institutions, level of cooperation and socio-economic variables to explain the way people voted in the referendum.

Since our dependent variable, voting for the UK staying in the EU or the UK leaving the EU, depicts mutually exclusive categories we use a Probit regression model to analyse voters' choices. Hence, $F_j$ is the cumulative distribution function of the standard normal.

The general form for a Probit model is:

$$y_i = \alpha + \beta X_i + \varepsilon_i \tag{9}$$

where $y_i$ is the voting choice, UK leaving the EU or UK staying in the EU, of voter $i$. Hence, the voting choice $y_i$ is a binary dependent variable with a value of 1 if the respondent voted to leave the EU and value of 0 if the respondent voted to stay in the EU. The parameter $\alpha$ is the model intercept that needs to be estimated; $\beta$ is a vector of parameters associated with the vector of independent variables $X_i$ consisting of the set of 6 personality traits, the level of trust in the UK government and the EC, the individual's level of cooperation and a set of socio-demographic variables including age, gender, level of education, income level and employment situation; $\varepsilon_i$ represents the model error term, where $\varepsilon \sim N(0, 1)$.

## 3. Results and Discussion

From a total of 599 respondents who could vote, 544 did vote in the 2016 referendum (90.8% of all respondents). This is higher than the 72.2% turnout of the 2016 referendum. This is probably due to the fact that people who answered the questionnaire were keen to participate in voting. Of these 544 respondents, 260 (47.8%) voted to remain in the EU and 284 (52.2%) voted to leave the EU. Table 2 shows the descriptive statistics of the variables used in the analysis (N = 530).

**Table 2.** Descriptive statistics of dependent variable and covariates used in the Probit model.

| Variables | Mean/% | Std. Dev. | Min | Max |
|---|---|---|---|---|
| Brexit_2016 | 52.97 | - | 0 | 1 |
| Agency | 2.30 | 0.68 | 1 | 4 |
| Agreeableness | 3.11 | 0.66 | 1 | 4 |
| Openness | 2.73 | 0.57 | 1 | 4 |
| Neuroticism | 2.22 | 0.68 | 1 | 4 |
| Extraversion | 2.69 | 0.67 | 1 | 4 |
| Conscientiousness | 3.14 | 0.59 | 1 | 4 |
| Age | 49.01 | 14.90 | 18 | 75 |
| Gender (female = 2) | 47.48 | - | 1 | 2 |
| Trust UK government | 2.39 | 1.26 | 1 | 5 |
| Trust EC | 2.60 | 1.28 | 1 | 5 |
| Cooperative | 45.95 | 32.94 | 0 | 100 |
| Not in paid employment | 13.08 | - | 0 | 1 |
| Working | 57.76 | - | 0 | 1 |
| Student | 2.43 | - | 0 | 1 |
| Retired | 20.37 | - | 0 | 1 |
| Unemployed | 6.36 | - | 0 | 1 |
| Income (<GBP 20k) | 38.88 | - | 0 | 1 |
| Income (GBP 20k–39k) | 37.76 | - | 0 | 1 |
| Income (GBP 40k–59k) | 14.58 | - | 0 | 1 |
| Income (GBP 60k and more) | 8.79 | - | 0 | 1 |
| Secondary school | 36.42 | - | 0 | 1 |
| College | 29.81 | - | 0 | 1 |
| University | 33.77 | - | 0 | 1 |

Min and max values indicate the minimum and maximum values that each variable can take.

We use the model results to calculate the predicted the probabilities of voting for the UK to leave the EU. Our sample predicted that probability to leave the EU was 52.97%, slightly above the outcome of 2016 referendum, 51.9%.

Table 3 shows the results obtained from the Probit regression analysis. The following personality traits, agency, openness to experience and extraversion were related to the way voters voted during the 2016 referendum. Voters who have self-confidence, forcefulness, assertiveness, outspokenness and dominance as personality characteristics (i.e., agency) were more likely to vote for the UK to leave the EU. This type of personality may resonate with a voter's perception about the UK being able to work autonomously and free from the control of the EU (i.e., EU regulations). Our results are consistent with Caprara and Zimbardo (2004), Fatke (2017) and Wang (2016), who found that voters' personality traits are associated with the ideology of their preferred political option. In this line, Forss and Magro (2016) pointed out that feelings of independence, nationalism and pride in the British way of life being highlighted by Brexit advocates played a role in both the Brexit debate and people's vote, along with facts-based arguments of experts. In addition, a belief of national greatness and feeling of national pride has been linked to the result of the Brexit referendum by de de Zavala et al. (2017) and to personality traits such as agreeableness and extraversion (Wang 2016). Wang (2016) found an indirect relationship between personality traits and voter choice in the United States presidential election of 2012. More specifically, higher levels of extraversion, conscientiousness and emotional stability were found to indirectly decrease the probability of voting for Obama (Wang 2016). Overall, different voters' personality traits seem to play a different role depending on the context (e.g., Brexit, US elections, Italian elections), as highlighted by Fatke (2017), who found that the effects of personality traits vary from country to country. Previously, Caprara et al. (1999) found a direct relationship between personality traits and the way people self-identified their preference in the 1994 Italian election (centre-right or centre-left). In particular, conscientiousness was associated with centre-right voters and agreeableness and openness with centre-right voters. Hence, voters' personality traits played a role in the Brexit vote. These may reflect voters' political ideology. Although voters for and against Brexit can be found in both major political parties in the UK, Alabrese et al. (2019) show that Labour supporters were more likely to support the UK remaining in the EU while Conservative supporters were more likely to support leaving the EU.

Focusing specifically on the associations found between individual personality traits, apart from agency, we also found that voters with an openness to experience trait are more likely to vote for the UK staying as part of the EU. Our result at the voter level on how openness is associated with voting for Brexit is in line with findings of Garretsen et al. (2018) at the regional level. Garretsen et al. (2018) point out the relevance of openness as a factor explaining voting choices at the UK county level and acknowledged that it is also a factor explaining voting choices at the individual level. Our results corroborate this; voter's closeness to experience, to forming part of a diverse community and the exchange of ideas and experiences are associated with voting for Brexit. Moreover, our results on openness support studies finding a positive relationship between fear of immigration and multiculturalism among voters in vulnerable situations (e.g., unemployed, poor, relatively low levels of education) (Abrams and Travaglino 2018; Goodwin and Heath 2017). On the effect of immigration on the UK 2016 referendum, Becker et al. (2017) found little evidence that exposure to the EU in terms of immigration and trade affected the referendum vote. However, it is worth pointing out that the authors captured real immigration levels at the regional level and not the views and perception on immigration at the individual level.

Extraverted individuals (i.e., individuals who are talkative, sociable, assertive, enterprising and energetic) were found to be more likely to vote for the UK to leave the EU. This suggests that extraversion, which can be linked to novelty seeking (i.e., individuals searching for novel experiences) (Gocłowska et al. 2019), played a role in the way UK voters voted in the referendum. A similar result was found by Barceló (2017) on the support for Catalan secession in the Spanish case.

**Table 3.** Determinants of voting for the UK leaving the EU.

| Variables | Mean | Std. Dev. |
|---|---|---|
| Intercept | 1.268 ** | 0.694 |
| Agency | 0.394 *** | 0.126 |
| Agreeableness | −0.014 | 0.126 |
| Openness | −0.359 ** | 0.158 |
| Neuroticism | 0.159 | 0.106 |
| Extraversion | 0.313 ** | 0.150 |
| Conscientiousness | −0.042 | 0.130 |
| Age | 0.002 | 0.006 |
| Male | | |
| Female | −0.158 | 0.136 |
| Trust UK government | 0.202 *** | 0.059 |
| Trust EC | −0.596 *** | 0.064 |
| Cooperative | −0.005 *** | 0.002 |
| Not in paid employment | | |
| Working | −0.341 | 0.212 |
| Student | −0.888 * | 0.473 |
| Retired | −0.508 ** | 0.255 |
| Unemployed | −0.539 * | 0.319 |
| Income (<GBP 20k) | | |
| Income (GBP 20k–39k) | 0.078 | 0.155 |
| Income (GBP 40k–59k) | −0.114 | 0.207 |
| Income (GBP 60k and more) | −0.150 | 0.239 |
| Secondary school | | |
| College | −0.325 ** | 0.157 |
| University | −0.751 *** | 0.161 |
| Log-likelihood | −273.283 | |
| Likelihood ratio test | 186.89 | |
| Pseudo R-squared | 0.255 | |
| N | 530 | |

*** Statistically significant at 1% significance level; ** statistically significant at 5% significance level; * statistically significant at 10% significance level.

Apart from an individual's personality traits such as agency, openness to experience and extraversion, an individual's level of trust in government institutions was found to be associated with the way individuals voted in the 2016 referendum. Voters who trust the UK government are more likely to vote for the UK to leave the EU, whereas voters trusting the EC are more likely to vote the UK to stay in the EU. Individuals more likely to vote for the UK to leave the EU seek ways to protect their own interest autonomously (Abrams and Travaglino 2018), favouring a system where they perceive the UK making its own policy decisions. Abrams and Travaglino (2018) point out the role played by UKIP in fostering mistrust against the EU and the British political establishment.

An individual's level of willingness to cooperate was found to be associated with being less likely to vote for the UK to leave the EU. This is an interesting result, since it suggests that people with a cooperative vision/pro-social behaviour (e.g., working together to achieve common objectives) were more likely to vote to remain in the EU than people who voted to exit the EU. People behave more pro-socially when they feel they belong to those who share a common identity (Georgiadis and Manning 2013). Hence, our results may indicate that Brexit voters (remain voters) feel a significantly stronger common identity as part of the UK (EU) than as part of the EU (UK). However, it is worth noting that pro-social behaviour can be affected by revealing information about own and others personality to individuals (Drouvelis and Georgantzis 2019) and interested parties may also influence voters' choice through affecting voters' pro-social behaviour.

Regarding how socio-demographic variables may be related with voters' choices, we found that level of education and employment situation were associated with the way they voted in the 2016 referendum. Voters with a higher level of education were

relatively less likely to vote for the UK to leave the EU. This result is in line with previous research, which also found an association between less educated voters voting for Brexit (Arnorsson and Zoega 2018; Alabrese et al. 2019). Furthermore, students, retired people and unemployed were relatively less likely to vote for the UK to leave the EU than people not in paid unemployment (not seeking work, e.g., houseman, housewife). These people may be receiving state benefits, which was found to be previously associated with being more likely to voting for the UK to leave the EU (Alabrese et al. 2019). We did not find any differences in the way people voted by age nor gender, holding everything else constant. There is no consensus in the literature on the association of age and gender with the way people voted (Alabrese et al. 2019; Liberini et al. 2019).

Table 4 shows the marginal effects of the explanatory variables on the probability of respondents voting for UK leaving the EU. Marginal effects measure the change in the probability of voting for UK leaving the EU with respect to a change in the value of an explanatory variable with everything else constant (i.e., $dydx$). The marginal effect of dummy variables (i.e., gender, employment situation, income and education) on the probability of voting for the UK to leave the EU measures the effect on this probability derived from a discrete change in the dummy variable (from 0 to 1). The benchmark groups for these variables are male, not in paid employment, with income less than GBP 20k and secondary school education.

**Table 4.** Marginal effects of explanatory variables on the probability of voting for UK leaving the EU.

| Variables | Mean | Std. Dev. |
|---|---|---|
| Agency | 0.115 *** | 0.035 |
| Agreeableness | −0.004 | 0.037 |
| Openness | −0.105 ** | 0.046 |
| Neuroticism | 0.046 | 0.031 |
| Extraversion | 0.091 ** | 0.043 |
| Conscientiousness | −0.012 | 0.038 |
| Age | 0.001 | 0.002 |
| Male | | |
| Female | −0.046 | 0.040 |
| Trust UK government | 0.059 *** | 0.017 |
| Trust EC | −0.174 *** | 0.014 |
| Cooperative | −0.002 *** | 0.001 |
| Not in paid employment | | |
| Working | −0.100 | 0.061 |
| Student | −0.259 * | 0.137 |
| Retired | −0.148 ** | 0.074 |
| Unemployed | −0.157 * | 0.092 |
| Income (<GBP 20k) | | |
| Income (GBP 20k–39k) | 0.023 | 0.045 |
| Income (GBP 40k–59k) | −0.033 | 0.060 |
| Income (GBP 60k and more) | −0.044 | 0.070 |
| Secondary school | | |
| College | −0.095 ** | 0.045 |
| University | −0.219 *** | 0.044 |

*** Statistically significant at 1% significance level; ** statistically significant at 5% significance level; * statistically significant at 10% significance level.

More interestingly, the probability of an individual who has a low level of agency (agency = 1) voting for the UK to leave the EU in the 2016 referendum is 33% compared to 77% for an individual with a high level of agency (agency = 4), given that all the rest of explanatory variables are set to their means. Likewise, the probability of an individual who has a low level of openness (openness = 1) voting for the UK to leave the EU in the 2016 referendum is 76% compared to 35% for an individual with a high level of openness (openness = 4), given that the rest of the explanatory variables are set to their means. Very extraverted individuals (extraversion = 4) were 68% likely to vote for the UK to leave the

EU in the 2016 referendum, compared to 32% of voters who were relatively less talkative, sociable, assertive, enterprising and energetic individuals (extraversion = 1). An association between voters' level of trust in government institutions, both UK government and the European Council, and voting for the UK to leave the EU was found. A voter trusting the UK government highly (trust UK government = 5) would be 73% likely to vote for the UK to leave the EU compared to 42% for a voter not trusting the UK government (trust UK government = 1). The association between the level of trust in the EU and voting for the UK to leave the EU is even stronger. A voter trusting the European Council highly (trust EC = 5) would be 9% likely to vote for the UK to leave the EU compared to 85% for a voter not trusting the UK government (trust EC = 1). Individuals with a low level of pro-social behaviour (cooperative = 0) were 62% likely to vote for the UK to leave the EU as opposed to 42% for individuals with a high level of pro-social behaviour (cooperative = 100), holding other variable values at their means. Differences in the likelihood to vote for the UK to leave the EU were found in voters' level of education and employment situation. For instance, voters with a university degree were 22% less likely to vote for the UK to leave the EU than voters with secondary education. Likewise, students, retired people and unemployed were 26%, 25% and 16% less likely to vote for the UK to leave the EU than voters not in paid employment (not seeking work, e.g., houseman, housewife).

The results above show the marginal effects of individual variables holding the other variables at their means. However, personality profile heterogeneity can be even greater than shown in the examples used above. For instance, the probability that an individual with a high level of agency (agency = 4), low level of openness (openness = 1) and high level of extraversion (extraversion = 4) voted the UK to leave the EU is 96% compared to 8% for an individual who has low level of agency (agency = 1), high level of openness (openness = 4) and low level of extraversion (extraversion = 1), holding the rest of variables at their mean values. If this individual had, in addition, a high level of trust in the UK government (trust UK government = 5) and a low level of trust in the EC (trust EC = 1), the probability for voting for the UK to leave the EU is 99.9% as opposed to 0.01% if the individual also had a low level of trust in the UK government (trust UK government = 1) and a high level of trust in the EC (trust EC = 5).

These results show how voters' heterogeneity beyond their socio-demographic characteristics matter to understand their vote choices. This may also have implications for parties interested in gaining voters' support. These parties may be willing to identify voters' personality traits in order to strategically target those who are likely to support their views and those who are less likely. Indeed, Big Data, including information obtained at the individual and household level through social media platforms and its use in elections, has been raised as an important issue in democratic societies. Thus, messages that may affect the confidence and trust that individuals have in institutions may have significant effects on the way people vote. Our results may suggest that messages, news and slogans that promote trust/distrust in government institutions (e.g., UK government, European Council) may be associated with how people vote. Notably, if voters' personality traits and socio-demographic variables could be identified (e.g., through internet surveys), then tailored messages might be delivered to targeted individuals to try to influence voting processes. Importantly, collecting this information by interested parties is relatively inexpensive. For instance, personality traits are moderately stable characteristics of individuals, so this information only needs to be collected once. Given the relative easiness in obtaining information on voters' personality traits and socio-demographic information, the use of targeted messages to individuals might affect the results of democratic processes.

## 4. Conclusions

We analysed the reasons for the outcome of the 2016 Brexit referendum using a different conceptualisation than previous analysis. By incorporating voters' personality traits, their level of trust in UK and European government institutions and their level of cooperation into the analysis, along with their socio-demographic characteristics, we have a

better understanding as to why Brexit happened. Our findings show that voters' heterogeneity goes beyond voters' socio-demographic characteristics. We show how grouping individuals into basic dimensions of personality based on their psychological profile (e.g., level of self-confidence, dominance, curiosity, ingenuity, talkative, sociable) in combination with trust in policy institutions and pro-social behaviour helps understanding the way individuals voted in the Brexit referendum. Consequently, identifying where that heterogeneity lies is important in both a priori (e.g., predicting voting outcomes) and a posteriori (e.g., understanding voting outcomes). As pointed out, vote choice research tends to focus mainly on the use of socio-demographic variables, disregarding relevant aspects such as personality profiles of voters. This study shows that accounting for other relevant variables when analysing individuals' vote preferences such as personality, trust in institutions and cooperative behaviour helps to better predict and explain individuals' voting decision-making. Although this study focuses on Brexit, its findings can be generalised to other contexts and voting choices (e.g., parliamentary elections). Hence, our results suggest that research to predict and/or explain voting choices/participation focussing only on socio-demographic variables may be limited and potentially lead to estimation issues related to omitted variable bias.

This study complements and extends previous work conducted by Garretsen et al. (2018) and May et al. (2021), where behavioural aspects were incorporated into the analysis of the 2016 Brexit referendum vote. We also complement and extend work by Alabrese et al. (2019), which accounts for individual level data, using the UK's largest household survey, Understanding Society, and focussing on individual socio-demographic characteristics. Individuals' trust in institutions (UK and EC) had an important role in the Brexit referendum outcome. Individuals' political trust (i.e., trust in the UK government and the EC) and intrinsic motives such as cooperative behaviour also played a role in the Brexit referendum outcome, which was also highlighted by Abrams and Travaglino (2018) and Liberini et al. (2019). Hence, based on what the results from this study suggest, we believe it would be beneficial to incorporate questions into already existing longitudinal surveys that enable profiling individuals according to their personality traits. In addition, it would also be advisable to include questions regarding individuals' level of trust in institutions and their pro-social behaviour.

Finally, we would like to emphasise how important is to take into account individual level information and depart from aggregated information in vote choice analysis whenever possible. In this way, more in-depth understanding of vote choices can be achieved.

**Funding:** This research received no external funding.

**Institutional Review Board Statement:** Not applicable.

**Informed Consent Statement:** Informed consent was obtained from all subjects involved in the study.

**Data Availability Statement:** The data presented in this study are available on request from the corresponding author.

**Conflicts of Interest:** The author declares no conflict of interest.

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
