# Peer review of "The Role of Personality Traits, Cooperative Behaviour and Trust in Governments on the Brexit Referendum Outcome"

_socsci, doi:10.3390/socsci10080309_

Round 1

Reviewer 1 Report

In this article, the authors aimed to identify predictors of voting on the 2016 Brexit referendum. They used a cross-sectional research and suggest that some types of personality, along with cooperation behavior, trust and some demographics were significant predictors of voting for leaving and remaining. This is a very interesting line of research, the identification of determinants that can play a role in decision making when it comes to political decision making. However, there are several concerns that support my decision of not recommending for publication. Please, find them below:

Abstract:

  • In the abstract, the authors write “We find voters’ choice was associated voters’ personality traits, their cooperative behaviour and their level of trust on government institutions along with socio-demographic characteristics such as voter’s level of education and employment situation”. The authors should describe the main results and not include all variables entered in the regression.

Introduction: 

  • Why were these indicators selected? What made the authors think of these variables that do not seem to fit together in the same theoretical framework? This is unclear to me. It is lacking to understand which research question were the authors trying to answer: which socio-demographics predicted voting for leaving/remaining in the UK, or which traits, or if cooperation, as a proxy for need for interdependence, was a predictor of voting. 
  • Why is this important? Why do we need to learn predictors of Brexit votes? How can this be extrapolated to other types of decision making when it comes to political interaction?
  • Conceptual framework should be read from left to right.
  • The authors should be careful when using the word ‘influence’ to describe the conceptual framework in Fig. 1, as influence implies causality.

Method:

  • When describing the variables/scales, give examples of items used, the anchors for the scales in which data were collected.
  • Level of trust in both UK and EU is not described in the measures.
  • Please, provide correlations between variables. Cooperation and personality traits are treated as independent variables, as well as trust in institutions. These should not be correlated with each other to be considered independent.
  • The sample is not described in the method section - only total N.

Results and discussion:

  • Descriptives such as gender, income, voting, should be expressed in percentage.
  • The authors should not describe and discuss relationships between variables that are not statistically significant (e.g., the positive relationship between neuroticism and voting).
  • The convencional p-value is 5% and not 10%. Results with p > .05 should not be flagged.

Conclusion:

  • The authors say that their findings have policy relevant insights. Which are those?
  • Sentences like these should be toned down: “Our results suggest that messages, news, slogans that promote trust/distrust in government institutions (e.g. UK government, European Council) can influence how people vote. Notably, if voters’ personality traits and socio-demographic variables can be identified (e.g. through internet surveys), then tailored messages can be delivered to targeted individuals to influence voting processes.”
  • These conclusions are not aligned with the research presented above. The authors identified predictors of voting for leaving EU - predictors, not factors - they should not use causal language, as this is not experimental research. Also, these conclusions address parties, with distinct political views, in a different context than the one that the UK voted for when voting for leaving or remaining in the EU. The message that I read here is that, based on identifying one’s personality traits, it is possible to send someone a message that is going to influence one’s vote in favor of something. These are not the findings here.

Reviewer 2 Report

The article addresses an interesting and relevant topic. However, there is a number of major shortcomings.

The introduction is misunderstood. It confuses the reader and fails to situate the research problem properly. Now we are facing a correct contextualization of the period studied, the referendum on Brexit. However, there is no justification for why it is relevant to know the personality traits of voters and their impact on voting. This section should be rewritten, better stating the research problem and improving the justification and advisability of studying it. Above all, the literature review on the role of people's personality and personal traits and how these come to influence their decision or political stance should be improved and expanded. In this regard, my question is why these personality traits and not different ones?

Considering the period analysed, the sample is considered to be limited (only 535 valid responses from the survey). Likewise, in the justification of the parameters to be analysed, it is not clear with what criteria the intensity of each type of personality is measured. Specifically, it is stated that the 6 individual level personality traits were calculated following Lachman and Weaver (1997), but the authors do not delve into explaining this technique.

The data provided in the tables allow interesting statements to be made, but, in general, the presentation of results is synthetic, very descriptive and has little depth. The analysis of the results obtained should be improved. The discussion of findings in relation to the previous literature is done only superficially. In addition, it is not clear what are the original and new contributions that this work makes to the field of social science. The authors fail to present generalizable results outside the case study.

Overall, my recommendation is to reject the paper.

Round 2

Reviewer 1 Report

I would like to thank to the authors the work they have done with the manuscript. There is just one thing that I have trouble incorporating: it is the discussion of relations that are not statistically significant. We use the rule of thumb of a p-value that is < .05, stating that these two variables are not associated by chance. When we discuss a relation between variables in which the p-value is > .05, we are discussing a non-existent relation.

And I do not see the added value of flagging p-values that do not represent meaningful relations between variables. 

Reviewer 2 Report

It is appreciated that the authors have applied all the comments and suggestions. Currently the manuscript presents a greater solidity since the justification of the topic presents a greater depth and the methodology is much clearer in its exposition and explanation.  It would be convenient to reinforce the conclusions, mainly, making a discussion with the previous literature and highlighting the findings found throughout the analysis of results.
